# Genetic Profile of Patients with Limb-Girdle Muscle Weakness in the Chilean Population

**DOI:** 10.3390/genes13061076

**Published:** 2022-06-16

**Authors:** Mathieu Cerino, Patricio González-Hormazábal, Mario Abaji, Sebastien Courrier, Francesca Puppo, Yves Mathieu, Alejandra Trangulao, Nicholas Earle, Claudia Castiglioni, Jorge Díaz, Mario Campero, Ricardo Hughes, Carmen Vargas, Rocío Cortés, Karin Kleinsteuber, Ignacio Acosta, J. Andoni Urtizberea, Nicolas Lévy, Marc Bartoli, Martin Krahn, Lilian Jara, Pablo Caviedes, Svetlana Gorokhova, Jorge A. Bevilacqua

**Affiliations:** 1Marseille Medical Genetics Université, INSERM, U 1251, Aix-Marseille Université, 13005 Marseille, France; mathieu.cerino@ap-hm.fr (M.C.); mario.abaji@ap-hm.fr (M.A.); sebastien.courrier@univ-amu.fr (S.C.); habana75@virgilio.it (F.P.); yves.mathieu@u-bourgogne.fr (Y.M.); nicolas.levy@univ-amu.fr (N.L.); marc.bartoli@univ-amu.fr (M.B.); martin.krahn@univ-amu.fr (M.K.); svetlana.gorokhova@univ-amu.fr (S.G.); 2Programa de Genética Humana, Instituto de Ciencias Biomédicas (ICBM), Facultad de Medicina, Universidad de Chile, Santiago 8380492, Chile; patriciogonzalez@uchile.cl (P.G.-H.); atrangulao@gmail.com (A.T.); ljara@uchile.cl (L.J.); 3Unidad Neuromuscular, Departamento Neurología y Neurocirugía, Hospital Clínico Universidad de Chile, Santiago 8380492, Chile; camperom@uchile.cl (M.C.); rhughesg@gmail.com (R.H.); 4Unidad de Patología Neuromuscular, Departamento de Neurología y Neurocirugía, Clínica Dávila, Santiago 8431657, Chile; nico.earle@gmail.com (N.E.); ignacio.acosta.c@gmail.com (I.A.); 5Departamento de Anatomía y Medicina Legal, Facultad de Medicina, Universidad de Chile, Santiago 8380456, Chile; 6Unidad de Neurología, Departamento de Pediatría, Clínica Las Condes, Santiago 7591047, Chile; ccastiglioni@clinicalascondes.cl (C.C.); rociitocortes@gmail.com (R.C.); karinkleinsteuber@gmail.com (K.K.); 7Centro de Imagenología, Hospital Clínico Universidad de Chile, Santiago 8380492, Chile; jdiazjara@gmail.com; 8Neurología Pediátrica Hospital Roberto del Río, Universidad de Chile, Santiago 8380492, Chile; carmenpazvargas@yahoo.com; 9Institut de Myologie, Hôpital de la Pitié-Salpêtrière, 75013 Paris, France; zberea@gmail.com; 10Department of Medical Genetics, Hôpital Timone Enfants, APHM, 13385 Marseille, France; 11Programa de Farmacología Molecular y Clínica, ICBM, Facultad de Medicina, Universidad de Chile, Santiago 8380492, Chile; pablo.caviedes@cicef.cl; 12Centro de Biotecnología y Bioingeniería (CeBiB), Departamento de Ingeniería Química, Biotecnología y Biomateriales, Facultad de Ciencias Físicas y Matemáticas, Universidad de Chile, Santiago 8380492, Chile

**Keywords:** limb-girdle muscle weakness, LGMD, hereditary myopathies, high-throughput sequencing, next-generation sequencing, Chile

## Abstract

Hereditary myopathies are a group of genetically determined muscle disorders comprising more than 300 entities. In Chile, there are no specific registries of the distinct forms of these myopathies. We now report the genetic findings of a series of Chilean patients presenting with limb-girdle muscle weakness of unknown etiology. Eighty-two patients were explored using high-throughput sequencing approaches with neuromuscular gene panels, establishing a definite genetic diagnosis in 49 patients (59.8%) and a highly probable genetic diagnosis in eight additional cases (9.8%). The most frequent causative genes identified were *DYSF* and *CAPN3*, accounting for 22% and 8.5% of the cases, respectively, followed by *DMD* (4.9%) and *RYR1* (4.9%). The remaining 17 causative genes were present in one or two cases only. Twelve novel variants were identified. Five patients (6.1%) carried a variant of uncertain significance in genes partially matching the clinical phenotype. Twenty patients (24.4%) did not carry a pathogenic or likely pathogenic variant in the phenotypically related genes, including five patients (6.1%) presenting an autoimmune neuromuscular disorder. The relative frequency of the different forms of myopathy in Chile is like that of other series reported from different regions of the world with perhaps a relatively higher incidence of dysferlinopathy.

## 1. Introduction

Hereditary myopathies comprise a large spectrum of degenerative muscular disorders genetically determined by several hundred pathogenic variants in distinct genes, with novel disease-causing mutations and genes being identified each year [1]. Despite growing interest and awareness regarding neuromuscular disorders in Latin America, the incidence and prevalence of hereditary myopathies is largely unknown on the continent [2]. In Chile, the number of patients affected with hereditary myopathies is approximately 6000 [3], but there are no specific registries to subcategorize distinct forms of myopathies.

Among these, limb-girdle muscular dystrophy (LGMD) is a broad and heterogeneous category of inherited muscle diseases involving proximal pelvic or scapular muscle weakness [4,5]. The clinical phenotype varies and overlaps widely from severe infantile or teenager onset forms to milder late-onset forms in which affected individuals have a slow progression and a relatively preserved lifespan [4,5]. Based on the inheritance pattern, LGMD is primarily divided into autosomal dominant (LGMD-D) and autosomal recessive (LGMD-R) forms [6]. LGMD-D roughly represents 10% of LGMDs and encompasses five subtypes (i.e., LGMD-D1 to D5), while the most frequent group, LGMD-R, comprises 24 recessive forms (LGMD-R1 to R24), each of which is caused by pathogenic variants in different genes [1,5,6]. The overall estimation of LGMD prevalence worldwide ranges from 1:14,500 to 1:123,000 [7]. However, the prevalence of LGMD and other forms of hereditary myopathy varies according to geographical areas [8] and to the diagnostic yield, the latter depending mainly on the selection criteria of patients and on the methodology used [9,10,11,12]. In spite of a consensus to delineate LGMD, the differential diagnosis of patients presenting with similar clinical features, namely, limb-girdle muscular weakness (LGMW) associated or not with elevated serum creatine phosphokinase (CK) levels, is wide, and obtaining a definitive and timely diagnosis for some rarer forms of LGMW remains challenging. Additionally, the recent update of the LGMD classification [6] led to the reconsideration of some forms of LGMD previously classified as such and no longer considered LGMD in the latest classification, as well as myopathies with LGMW previously not considered as LGMD and now included in this group [4,6]. Furthermore, novel causative genes, pathophysiological mechanisms and phenotypic presentations of neuromuscular disease are being continuously described [1,5].

High-throughput sequencing (HTS) has revolutionized the diagnosis of rare diseases, enabling individualized precision medicine. Targeted exome (or gene panel) allows for the evaluation of several genes simultaneously, improving the molecular diagnosis of Mendelian diseases, especially when they present heterogeneous phenotypes (i.e., LGMD and LGMW). Additionally, HTS has proven to increase the molecular diagnosis of patients with LGMW, since it generates more data at a lower cost, accelerating the process of identification of pathogenic variants and new genes associated with Mendelian diseases [13,14,15,16,17].

Here, we report the genetic findings of a series of Chilean patients presenting with limb-girdle muscle weakness (LGMW) of unknown etiology through clinical characterization, followed by a next-generation sequencing (NGS) panel approach. Since accessibility to a definitive molecular diagnosis was poor until recently in Chile, the main goal in this study was to develop the incidence profile of the different forms of hereditary myopathies in Chile.

## 2. Patients and Methods

### 2.1. Patient Enrollment

Enrollment of patients was undertaken between March 2015 and December 2019 at the Neuromuscular Unit, Department of Neurology and Neurosurgery, Hospital Clínico Universidad de Chile, which is one of the two university-based reference centers for adult neuromuscular disorders in the country. The patients who consented had to fulfill at least one of the following criteria: (1) clinical features of LGMD or LGMW with or without distal involvement; (2) increased plasma creatine kinase (CK) levels (this included cases of isolated hyperCKemia); (3) to be symptomatic or subsymptomatic relatives of known affected patients. Subjects excluded from the study were patients with clinical diagnostic features of myotonic dystrophy type 1 or 2 (DM-1 of DM-2); facioscapulohumeral dystrophy (FSHD), Duchenne muscular dystrophy (DMD) or oculopharyngeal muscular dystrophy (OPMD); as well as patients already genetically confirmed with LGMD and any other hereditary myopathies. The subjects were clinically assessed and characterized through a protocol comprising epidemiological data collection, force testing, muscle magnetic resonance imaging (MRI); clinical electrophysiology (NCV and EMG); histological and biological analyses of muscle and/or blood samples. DNA was extracted from peripheral blood or saliva and analyzed using different neuromuscular gene panels, according to availability at the time the genetic analysis (i.e., MyoPanel2 during 2015–2016; NGS-DLE during 2016; CL-NGS during 2017–2018; CNMDP during 2018–2019, see below). Further immunological testing for antimuscle-specific autoantibodies (MSAs), muscle-associated autoantibodies (MAAs), anti-HMGCR antibodies or other relevant serologic assays were also performed when possible in those cases for which HTS results were inconclusive (Appendix A).

Ninety-six patients were eligible for enrolment, eighty-two of whom were eventually included and completed this HTS study protocol (Figure 1). The 14 cases excluded corresponded to 5 patients that did not fully complete the required assessments; one family, comprising 3 affected individuals, was diagnosed with a *DOK7*-related CMS [18]; 3 additional patients were affected with desminopathy, which initiated their genetic study on a single-gene approach independently, earlier to the inception of this study [19]; another 3-member family that was affected with titinopathy studied through a different NGS protocol [12,20]. A detailed description of the ancillary testing findings was not the objective of this report, but it is discussed when phenotypic information is relevant to determine a variant’s pathogenicity (i.e., protein expression in muscle biopsy) (Appendix A).

### 2.2. Sequencing

High-throughput sequencing was performed with one of four different methods available at different times throughout the course of the protocol, depending on the patient’s clinical diagnostic hypothesis and the sequencing method availability. In eight cases (i.e., P5-Myo029; P17-Myo067; P81-Myo157; P28-Myo090; P32-Myo094; P59-Myo137; P61-Myo140; P82-Myo158), the diagnosis was established based on the clinical features and the absence of protein expression in the muscle biopsy, even though the molecular diagnosis could not be totally confirmed (i.e., P17-Myo067 with one *SGCG* mutation and the absence of γ-sarcoglycan in the muscle biopsy; P61-Myo140 with one pathogenic *DYSF* mutation and the absence of DYSF in the muscle biopsy) (Figure 1).

Appendix A depicts the complete list of the genes assessed in each NGS panel used. The HTS methodologies used were:MyoPanel2, consisting of 306 neuromuscular disease-causing genes designed at the Marseille Medical Genetics Institute (Aix-Marseille University, Marseille, France) [14]. The enrichment was performed using HaloPlex technology (Agilent Technologies^TM^), followed by sequencing on the NextSeq500 (Illumina^TM^) by Helixio^TM^ (Biopôle Clermont-Limagne, France). The bioinformatic analysis was performed as previously described [14];DLE-NGS—DLE Laboratory, Sao Paulo, Brazil, consisting of ten genes, the nine most-frequent LGMD-causing genes: *CAPN3*; *DYSF*; *SGCG*; *SGCA*; *SGCB*; *SGCD*; *FKRP*; *ANO5*; *TCAP* and *GAA*, as described elsewhere [10]. The coding regions and 10 nucleotides from the exon-intron junction from the included genes and intronic variants were customized with Agilent Sure-Select capture covering above 98% of target regions at 20x or greater. Nine genes and 154 corresponding exons related to muscular dystrophies and Pompe disease were included. Deep intronic variants were also investigated. Flanking exon/intron regions up to 25 base pairs (bp) were sequenced as well as known intronic variants if outside this range. The coding and flanking intronic regions were enriched using a Custom SureSelect QXT kit (Agilent technology) and were sequenced using the Illumina NextSeq 500 system. Only variants (SNVs/small indels) in the coding region and the flanking intronic regions (+10 bp) with a minor allele frequency (MAF) < 5% were evaluated. The ExAC, 1000Genomes and ABraOM projects were used to determine the frequency of the variants; a CADD score over 20 was the threshold to classify the in silico damaging prediction of the variant to the final protein, and other published information and laboratory databanks were used to further classify the variants. Patients who had pathogenic variants in homozygous or compound heterozygous state for *GAA* consistent with Pompe disease had alpha-glucosidase activity measured in the same paper filter card by fluorometry. After sequencing, the base call generated BCL files that were converted to FASTQ using the BCL2FASTQ script. The aligned file was then used for calling variants with the Samtools software, followed by annotation using the Variant Effect Predictor (VEP). “VCF” files annotated with VEP and in-house scripts were converted to tabulated tables and incorporated frequency information from variants already sequenced as well as Reactome and OMIM information. Quality analysis of the sequencing and call of variants was conducted by FASTQ and BAM files checked with Qualimap software. In addition, the average size of sequenced reads, aligned reads, transition rate, transversion, insertion, and deletion were surveyed. The nomenclature followed the HGVS guidelines [10];CL-NGS panel, set-up at the Instituto de Ciencias Biomédicas, Facultad de Medicina, Universidad de Chile, comprising 15 neuromuscular disease-associated genes including *LMNA*; *CAV3*; *DNAJB6*; *CAPN3*; *DYSF*; *SGCG*; *SGCA*; *SGCB*; *SGCD*; *FKRP*; *ANO5*; *FKTN*; *EMD*; *FHL1* and *DES*. Enrichment of coding regions by multiplex PCR and library preparation was performed with the Ion AmpliSeq Library Kit 2.0 (Thermo Fisher Scientific Inc., Waltham, MA, USA) according to instructions by the manufacturer. The pool of primers used for multiplex PCR was designed with the Ion Ampliseq Designer 5.6.3 tool (Thermo Fisher Scientific) covering 99% of the target regions. It amplified the targeted exons and >10 bp of surrounding intronic regions, with amplicons ranging from 125–375 bp. Emulsion PCR for clonal amplification of DNA in spheres was performed with the Ion PGM OT2 Hi-Q view kit (Thermo Fisher Scientific) on OneTouch 2 equipment. Sequencing was performed on an Ion Personal Genome Machine (Ion PGM system, Applied Biosystems) sequencer with the Ion sequencing kit, PGM Hi-Q view (Thermo Fisher Scientific), using the protocol provided by the manufacturer. The number of samples to be sequenced per run was calculated to achieve a minimum coverage depth of 100×. The variant calling was made with the Ion Reporter software (Thermo Fisher Scientific) using the default germline variant settings and hg19 as the genome reference;Invitae Comprehensive Neuromuscular Disorders Panel (CNMDP), comprised of 123 neuromuscular disease-causative genes including deletions and duplications (https://www.invitae.com/en/providers/test-catalog/test-03280, accessed on 1 April 2019). For the case of P53-Myo120, a panel of 137 genes was used. Genomic DNA obtained from the sample was enriched for targeted regions using a hybridization-based protocol and sequenced using Illumina technology. All targeted regions were sequenced with ≥50× depth or supplemented with additional analysis. Reads were aligned to a reference sequence (GRCh37/hg19), and sequence changes were identified and interpreted in the context of a single clinically relevant transcript. Enrichment and analysis focused on the coding sequence of the indicated transcripts, 20 bp of flanking intronic sequence and other specific genomic regions demonstrated to be causative of disease at the time of assay design. Promoters, untranslated regions and other noncoding regions were not otherwise interrogated. For some genes only, targeted loci were analyzed (Appendix A). Exonic deletions and duplications were called using an in-house algorithm that determined the copy number at each target by comparing the read depth for each target in the proband sequence with both mean read-depth and read-depth distribution, obtained from a set of clinical samples. *TTN* exons 45-46, 147, 149, 164 and 172-201 (NM_001267550.2) were excluded from analysis. This assay unambiguously detects SMN1 exon 8 copy number and sequence variants as well as sequence variants outside of exon 8, but this assay cannot determine whether the variant is in SMN1 or SMN2. CNVs of exons 1–6 of SMN1 or SMN2 are not reported. Confirmation technologies included any of the following: Sanger sequencing, Pacific Biosciences SMRT sequencing, MLPA and Array CGH.

### 2.3. DNA Extraction and Sample Collection

For the samples analyzed with MyoPanel2 and CL-NGS panels, DNA extraction was performed using the salting out method from peripheral blood; resuspended in buffer Tris-EDTA, BioUltra Sigma-Aldrich and quantified using a Nanodrop™ spectrophotometer, an approximate dilution of 20 ng/mL was obtained and confirmed through a fluorometric quantification in Qubit™. For the DLE-NGS panel, DBS samples were collected and sent to DLE laboratories in Sao Paulo, Brazil. For the Invitae^TM^ Comprehensive Neuromuscular Panel analysis, saliva or blood samples were collected by Genometrics (Santiago, Chile) and sent to Invitae laboratories in San Francisco, CA, USA.

### 2.4. Variant Analysis

All variants were classified according to the ACMG recommendations [21] adapted to LGMD genes [22]. Variants in *TTN* were classified according to the *TTN*-specific recommendations [23]. Briefly, we applied the PVS1 criteria according to ClinGen Sequence Variant Interpretation (SVI) recommendations for Interpreting the Loss of Function PVS1 ACMG/AMP Variant Criteria [24].

We applied a PP3 score (in silico prediction of pathogenicity) if the REVEL score was above 0.7. REVEL scores [25] were obtained from the UCSC browser. HSF [26], MaxENT [27] and SpliceAI [28] were used to assign PP3 for noncanonical splicing variants using the default thresholds. PM3 score (in trans with a pathogenic variant) was assigned according to the SVI Recommendation for in trans Criterion PM3 (Version 1.0, https://www.clinicalgenome.org/site/assets/files/3717/svi_proposal_for_pm3_criterion_-_version_1.pdf, accessed on 04 February 2022). The PP1 score (segregation data) was assigned according to the recommendations by the Hearing Loss ClinGen Working group that focused, in part, on recessive disorders [29]. The thresholds for the allele frequency criteria were as following: AR disorders: PM2—0.01%, BS1—0.1% and BA1—0.2%; AD disorders: PM2—0.004%, BS1—0.02% and BA1—0.2%. The POPMAX filtering allele frequencies were obtained from gnomAD v2.1.1 [30]. The PM1 criterion was not used, since the functional domains of the analyzed genes were not devoid of benign variation. The BP1/PP2 codes were not used, since both benign and pathogenic missense variants were present in most genes analyzed. PP5 and BP6 were not used following the recommendations by the ClinGen SVI Working Group [31].

## 3. Results

Demographic data are described in Table 1. Since this study was based in a neuromuscular clinical unit dedicated to adult patients, most patients (91.5%) were >18 years old with comparable percentages of men (57.3%) and women (42.7%) and a median age of 36.8 years (±13.9). All patients but three were Chilean (n = 79, 96.3%); P31-Myo093 was from Ecuador, P44-Myo106 was Bolivian and P51-Myo118 was Peruvian, all living in Santiago de Chile at the time the study was performed. Among the Chilean patients, 64.6% (n = 53) were from the Santiago Metropolitan area; 15.9% (n = 13) originated from the south (i.e., cities of Talca, Chillán, Concepción, Temuco and Puerto Montt); 8.5% (n = 7) from northern cities (i.e., Iquique, Antofagasta, Copiapó and La Serena) and 7.3% (n = 6) from the central area of Chile (i.e., Rancagua, Valparaíso and Viña del Mar). The time of disease duration at enrollment was highly variable, thus influencing the “time-to-diagnosis” that ranged from 3 to 50 years (mean = 11.2 years).

Figure 2 and Appendix A summarize the results of the NGS genetic screening. Pathogenic or likely pathogenic variants were identified in 49 of the 82 patients (59.8%) involving 21 different genes. This group included P70-Myo135 and P71-Myo138 for which *SMN1* analysis was completed with MLPA. In eight additional patients (9.8%), an almost-definite diagnosis was stablished if the patient had a phenotype closely matching that of the suspected mutated gene (i.e., P5-Myo029, *RYR1*; P28-Myo090, *VCP*) or had only one pathogenic variant or VUS of the two expected recessive variants on the affected gene accompanied by the absence of the target protein in the muscle (P17-Myo067 and P81-Myo157, *SGCG*; P32-Myo094; P59-Myo137; P61-Myo140; P82-Myo158, *DYSF*) (Figure 1 and Appendix A).

No disease-causative variants were identified in 25 cases (30.5%), including 5 cases (6.1%) that were considered partially solved, which corresponded to patients with VUS partially matching the phenotype (i.e., P6-Myo031 and P57-Myo141 (*TTN*); P42-Myo104 (*DYSF*); P62-Myo141 (*RYR1*); P65-Myo143 (*SYNE2*)). The relative frequency of the affected genes is shown in Figure 2. The most common causative genes identified were *DYSF* (18 out of 82 patients, 22.0%) and *CAPN3* (7 cases, 8.5%); *DMD* and *RYR1* followed with 4 (4.9%) cases each. *ANO5*, *SGCG*, *SNM1*; *PGYM*, *SCN4A*, *GAA*, and *VCP* presented with 2 cases (2.4% each). All the other genes were represented by 1 case (1.2% each).

*DYSF* (NM_003494.3) was the most mutated gene in this series, presenting with 11 pathogenic variants and two VUS. Of these, the recurrent pathogenic variant c.4887-2A > G in intron 44; as well as the VUS 5’UTR c.-116delG and c.1186G > A in exon 14 were novel. The VUS in 5’UTR was identified in a homozygous state in patient P32-Myo94, accompanied by the absence of dysferlin in the muscle biopsy. Two cases presented with a myalgia-hiperCKemia syndrome: patient P42-Myo104 carrying a homozygous VUS c.1186A > G in exon 14 of the *DYSF* gene and patient P33-Myo095 with a heterozygous recurrent pathogenic variant c.5979dupA on exon 53. The variants in exon 53: c.5979dupA (n = 10), exon 27: c.2858dupT (n = 9) and exon 26: c.2779delG (n = 5) were recurrently identified, representing altogether 64.1% of the total variants found in *DYSF* (i.e., 26.3%, 24.3% and 13.5%, respectively).

LGMD-R1 calpain 3-related was the second-most frequent diagnosis in our series. Seven pathogenic *CAPN3* variants were identified in seven patients. The novel mutation, c.107delG (p.Gly36ValfsTer21), was present in three out of seven unrelated patients from different regions of the country, and the mutation c.2362_2363delinsTCATCT (p.Arg788SerfsTer14) recurred in five out of the seven identified patients, all with a clear LGMD-R1 phenotype [32].

The *DMD* had different small deletions in patients P60-Myo139 and P75-Myo151, and the same frameshift deletion c.40_41del in *DMD* in P25-Myo086 and P58-Myo132, all of whom presented a Becker muscular dystrophy phenotype.

Variants in the *RYR1* gene were found in five patients including one VUS (i.e., P62-Myo141). Patient P5-Myo029 carried one *RYR1* and one *MYH7* pathogenic mutations. This patient presented with LGMW, hyperCKemia and myalgias with mild nonspecific muscle biopsy findings and muscle MRI within normal range and no cardiac alterations; these manifestations were considered within the spectrum of *RYR1*-related phenotypes. Patient P30-Myo092 harbored two pathogenic variants in *RYR1*, the novel c.4455_4459dup (p.Lys1487ThrfsTer16) in combination with the c.6502G > A (p.Val2168Met) variant. They presented with a retractile congenital myopathy and displayed histological and MRI findings compatible with an *RYR1*-related myopathy. Two siblings, P63-Myo142 and P64-Myo142.1, presenting with lower leg distal myopathy, with biopsy and MRI findings matching the genotype, were heterozygous for *RYR1*:NM000540.2: c.14209C > T; p.(Arg4737Trp). Patient P62-Myo141 carried the intronic *RYR1* VUS c.1577-5C > G (intron 14). They had a long-standing history of mild generalized weakness, myalgias and cramps with nonspecific histological manifestations, suggesting a congenital myopathy (i.e., variability in fiber size, type 1 fiber predominance) in line with the variable spectrum of *RYR1*-related phenotypes.

Two unrelated patients were affected with anoctamin-5-related LGMD-R12. P3-Myo017, descending from a consanguineous marriage, carried the novel *ANO5* c.2201T > C (p.Leu734Pro) mutation in the homozygous state. Interestingly, the second patient affected with LGMD-R12, P49-Myo116, was homozygous for the same novel mutation, in combination with the heterozygous c.191_192insA (p.Asn64LysfsTer15) pathogenic variant. Both unrelated cases showed phenotypic manifestations usually observed in anoctaminopathies.

Patient P9-Myo052 harbored two novel variants in exon 4 of the *FKRP* gene, c.919T > G; p.(Tyr307Asp) and c.877A > C; p.(Thr293Pro) in compound heterozygosity. Their clinical, imaging and biopsy findings were compatible with LGMD-R9.

Subject P66-Myo144 carried the novel heterozygous mutation c.709G > A (p.Glu237Lys) in *TPM3* gene. They had clinical and MRI features, and particularly histological findings, revealing a congenital fiber-type disproportion associated with cap myopathy, which are manifestations attributable to *TMP3* mutations, and was therefore classified as likely pathogenic according to the ACMG criteria [33].

Only one of the two causative mutations expected in *SGCG* were found in P17-Myo067 and P81-Myo157, both females that displayed an LGMD phenotype. In P17-Myo067, there was a heterozygous deletion of an entire coding sequence in *SGCG*, while P81-Myo157 carried the novel heterozygous VUS variant c.817T > A in exon 8 of *SGCG.* This latter sequence change replaces tyrosine with asparagine at codon 273 of the SGCG protein (p.Tyr273Asn), and the algorithms to predict its effect on protein structure and function strongly suggest that is disruptive. The variant had no frequency in the population databases (gnomAD v2.1.1, v3.1.2) and has not been reported in the literature in individuals with SGCG-related conditions before. In these two patients, SGCG was repeatedly absent in the muscle biopsy by immunohistochemistry, with normal expression of the other sarcoglycans and sarcolemmal proteins. Taken together, these findings were considered sufficient to establish the diagnosis on firm grounds.

Patients P70-Myo138 and P71-Myo138, who did not show causative mutations in the 15-gene CL-NGS panel, underwent multiplex-ligation probe amplification (MLPA) analysis for *SMN1* due to the clinical and epidemiological evidence obtained in a subsequent clinical reassessment. Both patients exhibited LGMD-like symptoms and signs showed a deletion in *SMN1*; consequently, a diagnosis of spinal muscular amyotrophy type 3 (SMA 3) was established. 

Cases P6-Myo031 and P57-Myo131, harboring, respectively, the variants c.94507G > A (p.Ala31503Thr) in exon 340 and c.70897dup (p.Leu23633ProfsTer17) in exon 326 of *TTN*, showed clinical, imaging and biopsy findings roughly compatible with titinopathy. The variant in exon 326 is already known as pathogenic and related to centronuclear myopathy and dilated cardiomyopathy [34]. However, the diagnosis of titinopathy could not be established in this family due to the poor phenotypic correlation, and the fact that several recent reports suggest that a single LOF variant in *TTN* should be, in combination with a missense pathogenic variant in exons 362-364, considered causative [23].

Patient P27-Myo089 developed clinical and electrophysiological features of Lambert–Eaton Myasthenic Syndrome (LEMS), with raised titers of anti-P/Q-type calcium channel antibodies. This woman also showed two pathogenic variants: the mutation c.95195C > T (p.Pro31732Leu) in the *TTN* gene and the mutation c.1847G > A (p.Trp616Ter) in *CACNA1S.* However, regarding the *TTN* variant, this patient did not show features of HMERF and had no family history of myopathy. Further, her muscle MRI and biopsy proved inconsistent with titinopathy. Conversely, loss-of-function variants in *CACNA1S* have not been reported to confer risk for autosomal dominant conditions. Lastly, the patient responded to a combined treatment with 3,4-diaminopyridine and pyridostigmine, thus behaving as a genuine LEMS.

Another novel likely pathogenic variant of the *VCP* gene (heterozygous variant c.648A > G (p.Ile216Met)) was found in P28-Myo090. According to the ACMG criteria, this mutation scored as a VUS. The clinical features of the patient, plus a biopsy that complied with histopathological criteria for IBM and clinical features together with compatible muscle MRI findings, suggested strongly that the mutation was causative. In addition, P56-Myo128, who presented an LGMW phenotype compatible with collagen VI-related myopathy, harbored the novel homozygous mutation c.4899del, p.Glu1634ArgfsTer32 in *COL6A3*. All the other causative mutations identified are known pathogenic variants that matched the phenotypic manifestations of the patients.

The other patients without identified causative mutations represent a heterogeneous group. Subjects P10-Myo053, P16-Myo064 and P50-Myo117 had increased serum anti- HMGCR antibodies, and P13-Myo056, elevated anti-SRP antibodies. All such cases were associated with clinical and/or additional features compatible with the diagnosis of immune-mediated necrotizing myopathy (IMNM). Cases P44-Myo106 and P53-Myo120 had clinical dystrophinopathy features and reduced expression of dystrophin in the muscle biopsy. Yet, they did not show *DMD* deleterious variations with the NGS panels; targeted *DMD* sequencing was also performed by Sanger in P44-Myo106, but it was negative, thus suggesting a deep-intronic mutation. Subject P20-Myo075 showed the *DYSF* VUS c.383G > A and c.2948A > C in a compound heterozygous state in a 10-gene NGS panel. She had insidious LGMW, myalgia and hyperCKemia together with a nonspecific muscle biopsy with normal expression of DYSF by IHC. Yet, she manifested clinical signs of motoneuron disease, fulfilling the diagnostic criteria for ALS through a disease course of five years, in turn, suggesting the *DYSF* variants were not pathogenic. The remaining cases, although having VUS in some (not shown), showed no plausible causative mutations (Table 2).

## 4. Discussion

In the present study, we describe a series of Chilean patients suffering from limb-girdle muscular weakness (LGMW) of unknown etiology, collected through a 57 month period between March 2015 and December 2019 and explored by high-throughput sequencing (HTS). To our knowledge, this type of diagnostic approach for patients with LGMW has not been implemented in Chile before. Disease duration at enrollment was highly variable, thus influencing the “time-to-diagnosis” that ranged from 3 to 50 years (mean = 11.3 years).

The overall conclusive diagnostic yield in this series was 59.8%, achieving a genetic diagnosis in 49 out of 82 patients who carried pathogenic or likely pathogenic variants, comprising 21 known neuromuscular disease-causing genes including 12 novel variants. This diagnostic yield is at the top of the reported in the literature for similar HTS testing approaches in LGMW diagnosis [9,10,11,12,13,35]. This can be partly due to the flexibility of the strategy implemented which allowed us to modify the study’s algorithm according to the results that were being obtained, a careful pretesting selection of patients, the large number of genes included at the end in the analysis, as well as a post-testing reanalysis in some cases that allowed to confirm the pathogenicity of the variants found (i.e., P28-Myo90 or P66-Myo144). The diagnostic yield reached 69.6% when an additional 9.8% were added by including eight patients that, although having a consistent genotype–phenotype correlate, failed to fulfill the strict criteria for pathogenicity. Moreover, the general diagnostic yield produced by the study was greater (76.8%; 63 out of 82 patients) if non-Mendelian illnesses were included (i.e., autoimmune disease, ALS). To us, this is the most relevant data, as our goal was to outline the differential diagnostic spectrum of patients with challenging LGMW syndromes.

As in other series, the most frequently affected genes were *DYSF* and *CAPN3* followed by *DMD* and *RYR1* [9,10,11,16,36,37]. All the other genes were represented with two or one case each including some genes rarely found even in large series of patients (*MTM1*, *GAA*, *VCP*, etc.), whereas other more commonly found genes (sarcoglycans, *TTN*, etc.) were seemingly underrepresented. Similar studies in Latin America are limited. In the series reported by Bevilacqua et al. [10], nine LGMD-causing genes and the *GAA* gene were screened and the inclusion criteria were less specific; the diagnostic yield of this study was 16%. Recently, by using the same ten-gene NGS panel approach, Schiava et al. [35] reported the results of an Argentinian series of 472 LGMD patients, with a diagnostic yield of 10.8%. Winkler et al. [36] studied a Brazilian cohort of 51 patients with a panel of 39 myopathy-causing genes, reaching a diagnostic yield of 60.8% when including cases with candidate variants. In line with our findings, in these three series, *DYSF* and *CAPN3* were found among the most frequently affected genes. Furthermore, in the cohort reported by Nallamilli and coworkers [9], comprising 4656 United States of America patients studied with an NGS panel of 35 LGMD-causing genes, the diagnostic yield was 27% and *CAPN3*, *DYSF*, *FKRP* and *ANO5* were the most affected genes. Similarly, in the series reported by Töpf et al. [11], 1001 patients affected by LGMW from Europe, the Middle East and North Africa were screened by whole-exome sequencing for 429 NMD-causing genes. Pathogenic variants were found in 52% of the patients; *CAPN3*, *RYR1*, *DYSF*, *ANO5*, *DMD* and *TTN* were the most affected genes. In this latter study, the inclusion criteria and the number of analyzed genes matched the ones we used, perhaps with a higher diagnostic difficulty, as most of the cases from European countries were analyzed after prescreening testing. In the Spanish series reported by González-Quereda et al. [12], the diagnostic yield was 43.9%, with *TTN* and *RYR1* being the most affected genes. However, in this cohort, the 207 patients comprised a wider spectrum of phenotypic presentations than in our study (i.e., children and adults; congenital myasthenic syndromes; congenital dystrophies) and were studied with a 116 NMD-disease-causing gene NGS panel. In a Chinese study conducted by Yue et al. [38] between 2013 and 2015, in which 180 Chinese patients suspected of LGMD were analyzed with an NGS panel covering 420 neuromuscular disease-causing genes, they achieved a positive diagnostic rate of 68.3%, the most common genes causing LGMD being *DYSF* (49.5%) and *CAPN3* (24.8%). Interestingly, in this study, which had a similar design to the one reported here, the frequency of the affected genes matched the Chilean one, showing a larger proportion of LGMD-R2 dysferlin-related cases (see below). There was no coincidence between the variants found in the most affected genes of the Chilean patients with the other reported series. For example, only one of the variants found in *CAPN3* was also identified in the Chinese series [38], similarly, only one variant in *CAPN3* coincided with the Argentine series [35] and none with the Brazilian one [36].

These differences in the diagnostic yield most probably are explained by the differences in the selection criteria, pretesting and post-testing completeness of the diagnostic algorithm which enabled to establish the plausibility of the variants found as well as the disparities of the classification criteria, the number of genes included in the panels and the methodology used in each study.

Even though larger studies are still warranted, we believe the data here reported reflect the prevalence of these forms of hereditary myopathies in Chile. Nevertheless, we acknowledge the series is most representative of the diversity of the adult Chilean population, including patients from neighboring countries grossly reflecting the distribution of the population in the country.

Furthermore, the design of our study had several methodological limitations. Large copy number variants were not systematically searched in the data sequencing. Diagnosis of certain patients could have also been overlooked due to the small size of the gene panel (i.e., 10 or 15 genes). In addition, we did not have the possibility to perform a complete segregation analysis in several patients, thus precluding us from validating the segregation criteria required in some cases to assess pathogenicity. Moreover, in consideration of the genes identified in our series, we know that mutations in some genes underrepresented or are absent in this series (i.e., *DES*, *TTN* and *LMNA A/C*) have been identified in the Chilean population [19,20]; thus, our data are incomplete. Likewise, since most patients were adults, causative genes determining early-onset myopathies of common occurrence (congenital myopathies, congenital dystrophies, etc.) were largely not observed. Inclusion of those cases in a series like this one, will clearly alter the statistical conclusions and must be considered in future epidemiological analysis of NMD in the country.

Even though in subjects P70 and P71, suffering from SMA, the causative mutations were missed by the HTS panel used, we retained these cases in the series, as late-onset SMA is a frequent and relevant differential diagnosis in LGMW. It also highlights the need to be aware of the limitations of the HTS panel approach for large deletions.

Although 25 out of 82 patients (20.5%) did not reach a genetic diagnosis, including 5 patients (6.1%) that were considered partially solved, a definitive conclusion of acquired autoimmune conditions was reached for 5 (6.1%) other patients in this subset, including case P50-Myo117, an 8 year old showing an LGMW phenotype with compatible muscle MRI and biopsy findings that revealed anti-HMGCR-antibodies but that was never exposed to statins [39]. This number of patients with autoimmune NMD is somehow expected in a series of adult individuals with LGMW of unknown cause [40], since autoimmune neuromuscular disorders may mimic LGMD [41]. Recognition of these patients is evidently important, since they constitute potentially treatable conditions.

In the cases of P44-Myo106 and P53-Myo120, with a dystrophinopathy phenotype and reduced expression of dystrophin in muscle biopsy but without pathogenic mutations identified in *DMD* through the HTS panels used, a deeper study of the *DMD* gene is required [42]. In these subjects, the plausibility of other causative genes is clearly hampered by the absence of mutations in all the other genes screened as part of the HTS panels performed. Similarly, some cases, such as P33-Myo095 and P42-Myo104, carrying only one pathogenic mutation or VUS in *DYSF*, detected through small, targeted panels and showing phenotypes compatible with a symptomatic carrier or atypical dysferlinopathy, would benefit from more thorough phenotyping and genotyping [43].

Several important conclusions can be drawn from our series. First, some forms of LGMD, like calpainopathies and sarcoglycanopathies are surprisingly less frequent in Chile compared with other reported series from North America, European and Latin American countries [9,10,11,12,35]. In contrast, dysferlinopathies seem to be relatively more frequent, similar to the Chinese series [38]. The Chilean dysferlinopathy database precedes and outnumbers the patients reported as part of this study [44,45], thus suggesting that the prevalence of dysferlinopathy is even higher than the 22.0% observed in this study and that other forms of LGMD are relatively less frequent in Chile than in other regions of the world [7,8,9,10,11,12,35]. We underline the identification of novel pathogenic variants in *DYSF*, *CAPN3*, *ANO5*, *SGCG*, *VCP*, *COL6A3* and *RYR1*, all bearing a plausible phenotypic correlation. Interestingly, recurrence of some of these variants in the series (i.e., *CAPN3*, *ANO5*, *PYGM* and *DYSF)* may be suggestive of a high level of consanguinity, a founder effect or a combination of both in the Chilean population.

Taken together, these data provide an estimate of the relative frequency of genes causative of LGMW in Chile. We validated the use of HTS testing as a very useful screening tool, both for positive identification of pathogenic variants and to exclude candidate genes, but in combination with a thoughtful preselection of patients as well as a rigorous post-sequencing analysis of plausibility to reach a satisfactory diagnosis [43]. In the same way that antibody testing appears to be necessary in those patients with LGMW of unknown genetic cause, it seems advisable to perform HTS screening in patients with myopathies of seemingly autoimmune basis, particularly if the disease course and/or response to treatment are not the expected ones [39].

## Figures and Tables

**Figure 1 genes-13-01076-f001:**
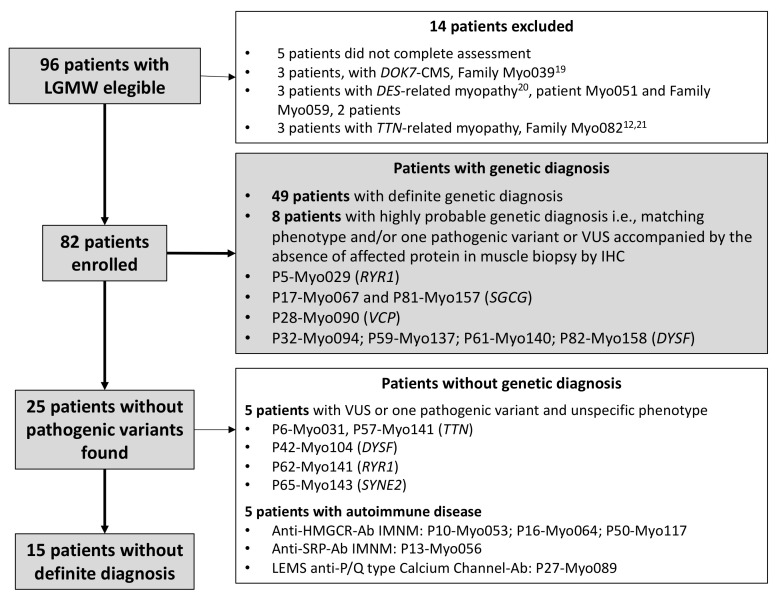
**Patient selection strategy and study design.** Grey boxes indicate the main flow of the protocol and patients with definite or highly probable genetic diagnosis. Patients excluded from the protocol or with no genetic diagnosis are shown in the white boxes. See also Table 1.

**Figure 2 genes-13-01076-f002:**
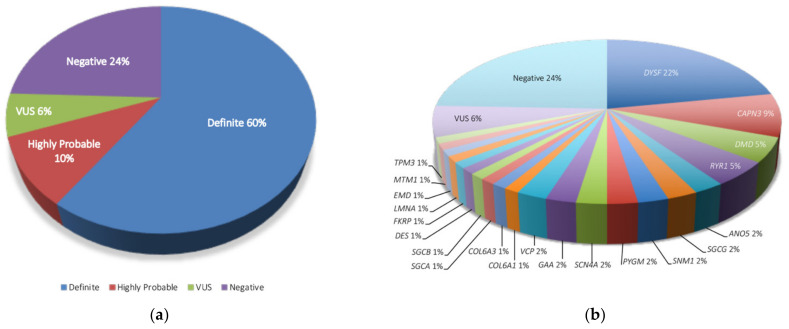
**Diagnostic yield of the study:** (**a**) percentage of patients with a definite genetic diagnosis, highly probable genetic diagnosis, negative diagnosis or variants of unknown significance (VUS). (**b**) number of patients with a definite diagnosis by gene, VUS or absence of causative mutation (i.e., negative).

**Table 1 genes-13-01076-t001:** Demographic data of the series.

Parameter	Statistics
Total, n	82
Female/male, n (%)	35 (42.7)/47 (57.3)
Age (y), mean ± SD (minimum–maximum)	36.8 ± 13.9 (8–68)
<18 years of age, n (%)	7 (8.5)
≥18 years of age, n (%)	75 (91.5)
Patients with definite genetic diagnosis, n (%)	49 (59.8)
Patients with highly probable genetic diagnosis, n (%)	8 (9.8)
Patients without genetic diagnosis, n (%)	25 (30.5)
Patients with any * conclusive diagnosis, n (%)	63 (76.8)
Patients without conclusive diagnosis, n (%)	19 (23.2)
Time from onset of diagnosis (y), mean ± SD (minimum–maximum)	11.2 ± 11.3 (3–50)
**Geographic Origin ****	**n (%)**
Northern Chile	7 (8.5)
Central Chile	6 (7.3)
Southern Chile	13 (15.9)
Santiago Metropolitan Area	53 (64.6)
Ecuador	1 (1.2)
Bolivia	1 (1.2)
Peru	1 (1.2)

* Any conclusive diagnosis implies genetic, serological or clinical diagnosis (i.e., P20-Myo075, diagnosed with ALS and patients with autoimmune NMD). ** Zones of Chile were roughly defined arbitrarily with reference to the Santiago Metropolitan Area. Patients originated from different cities in each case. Northern Chile: Iquique, Antofagasta, La Serena, Copiapó; Central Chile: Valparaíso-Viña del Mar, Rancagua; Southern Chile, Talca, Chillán, Concepción, Temuco, Puerto Montt. Patients from other countries lived in the Santiago Metropolitan Area at the time of the assessment.

**Table 2 genes-13-01076-t002:** Number of patients with definite genetic diagnosis by gene.

Gene	n, Affected Patients (% over 82 Samples)
*DYSF*	18 (22)
*CAPN3*	7 (8.5)
*DMD*	4 (4.9)
*RYR1*	
*ANO5*	2 (2.4)
*SGCG*	
*SNM1*	
*PGYM*	
*SCN4A*	
*GAA*	
*VCP*	
*COL6A1*	1 (1.2)
*COL6A3*	
*SGCA*	
*SGCB*	
*DES*	
*FKRP*	
*LMNA*	
*EMD*	
*MTM1*	
*TPM3*	
**VUS**	**5 (6.1)**
**Negative**	**20 (24.4)**
**Total**	**82 (100)**

## Data Availability

The data presented in this study are available on request from the corresponding author.

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
