# Peer review of "Genetic Profile of Patients with Limb-Girdle Muscle Weakness in the Chilean Population"

_genes, 2022, doi:10.3390/genes13061076_

Round 1
Reviewer 1 Report
The authors describe a series of Chilean patients suffering from limb-girdle muscular weakness (LGMW) of unknown etiology, explored by high throughput sequencing (HTS). This appears to be the first time this type of diagnostic approach for patients with LGMW has been implemented in Chile.
This diagnostic yield is at the top of the reported in the literature for similar HTS testing approaches in LGMW diagnosis. DYSF, CAPN3, and RYR1 were found the most frequently affected genes.
This is a clearly presented report that will be of interest to both physicians and researchers in the region and elsewhere.
Author Response
We thank reviewer 1 for the Report and approving comments on our work.
A complete grammar and spelling check has been performed, and amendments made accordingly, some phrases and paragraphs have been changed for clearness. Changes are indicated in the Highlighted R1 version of the manuscript. Also, the VUS variant reported for patient P32-Myo094 has been corrected.
Sincerely
Reviewer 2 Report
This is a study on genetically unsolved LGMD cases in a cohort of Chilenian patients. It provides important epidemiological data from South America, though not being representative. The presentation of the work is very well done, 82 patients are presented in detail and the genetic background could be clearly identified in a high percentage of cases.
I have only some minor points to be adressed:
1. The collection of the patients should be presented in the methods section, e.g. single-center study, how many of these highly specialized centers exist in Chile etc.
2. It would be nice to have an extension on the worldwide data on unsolved LGMD cases. It is hoghly important to understand which genes can be found ion certain areas, therefore a comparison with other countries (at least for the most frequently found mutations) would be much of an improvement. You did not mention some of the literature that was published in recent years in China, Germany etc., mostly exom based analysis of unsolved LGMDs.
Author Response
We thank Reviewer 2 for the constructive observations made as well as the favorable report.
A complete grammar and spelling check has been performed, and amendments made accordingly, some phrases and paragraphs have been changed for clearness. Changes are indicated in the Highlighted R1 version of the manuscript. Also, the VUS variant reported for patient P32-Myo094 has been corrected.
With regard to the specific comments made:
Point 1. "The collection of the patients should be presented in the methods section, e.g. single-center study, how many of these highly specialized centers exist in Chile, etc."
Response 1: There are no formally established highly specialized neuromuscular centers in Chile for adult patients; however, at the University of Chile Hospital (i.e. Us) and the Catholic University Hospital (an Education Ministry-depending institution and a private institution respectively), there are neuromuscular units where patients are referred unsystematically from all the country. Likewise, for pediatric patients, in the Public Health system, the Hospital Dr.Luis Calvo Mackenna and the Hospital Clínico San Borja Arriarán function as a reference center but again, no specialized NM units are implemented. All these University and Hospitals are at Santiago Metropolitan Region, the country's capital.
Nevertheless, the following sentence has been added to Section 2.1 Patient Enrollment:
"Enrollment of patients was undertaken between March 2015 and December 2019, ... at the Neuromuscular Unit, Department of Neurology and Neurosurgery, Hospital Clínico Universidad de Chile which is one of the two university-based reference centers for adult neuromuscular disorders in the country"
Point 2. "It would be nice to have an extension on the worldwide data on unsolved LGMD cases. It is hoghly important to understand which genes can be found ion certain areas, therefore a comparison with other countries (at least for the most frequently found mutations) would be much of an improvement. You did not mention some of the literature that was published in recent years in China, Germany, etc., mostly exon-based analysis of unsolved LGMDs."
Response 2. We appreciate this observation since allowed us to enlarge the comparison of our results and to include a just-published Latin American Reference. To respond to the reviewer's comment as well as to complete our discussion, two more references have been included and commented on in the manuscript (References 35 and 37, an Argentine and a Chinese NGS study on NM patients respectively). In Section 4. Discussion. The following paragraph, on page 12 lines 450-477, has been modified concerning this point:
"Recently, by using the same ten-gene NGS panel approach, Schiava et al. [35] reported the results of an Argentinian series of 472 LGMD patients, with a diagnostic yield of 10.8%. Winkler et al. [36] studied a Brazilian cohort of 51 patients with a panel of 39 myopathy-causing genes, reaching a diagnostic yield of 60.8% when including cases with candidate variants. In line with our findings, in these three series, DYSF and CAPN3 were found among the most frequently affected genes. Furthermore, in the cohort reported by Nallamilli and coworkers [9], comprising 4656 United States of America patients studied with a NGS panel of 35 LGMD-causing genes, the diagnostic yield was 27% and CAPN3, DYSF, FKRP, and ANO5 were the most affected genes. Similarly, in the series reported by Töpf et al [11], 1001 patients affected by LGMW from Europe, the Middle East, and North Africa were screened by whole-exome sequencing for 429 NMD-causing genes. Pathogenic variants were found in 52% of the patients, CAPN3, RYR1, DYSF, ANO5, DMD and TTN were the most affected genes. In this latter study, the inclusion criteria, and the number of analyzed genes matched the one we used, perhaps with a higher diagnostic difficulty, as most of the cases from European countries were analyzed after prescreening testing. In the Spanish series reported by González-Quereda et al. [12] the diagnostic yield was 43.9% being TTN and RYR1 the most affected genes. However, in this cohort, the 207 patients comprised a wider spectrum of phenotypic presentations than in our study (i.e., children and adults; congenital myasthenic syndromes; congenital dystrophies) and were studied with a 116 NMD-disease-causing gene NGS panel. In the Chinese study conducted by Yue et al [38] between 2013 and 2015, in which 180 Chinese patients suspected of LGMD were analyzed with a NGS panel covering 420 neuromuscular disease-causing genes, they achieved a positive diagnostic rate of 68.3%, being the most common genes causing LGMD DYSF (49.5%) and CAPN3 (24.8%). Interestingly, in this study that had a similar design to the one reported here, the frequency of the affected genes matched the Chilean one, showing a larger proportion of LGMD-R2 dysferlin-related cases (See below). There was no coincidence between the variants found in the most affected genes of the Chilean patients with the other reported series. For example, only one of the variants found in CAPN3 was also identified in the Chinese series [38], similarly, only one variant in CAPN3 coincided with the Argentine series [35] and none with the Brazilian one [36]."
Yours sincerely